# Impacts of Assimilating ATMS Radiances on Heavy Rainfall Forecast in RMAPS-ST

**Yanhui Xie [1],\*, Min Chen [1], Jiancheng Shi [2,3] , Shuiyong Fan [1], Jing He [1] and Youjun Dou [1]**

1 Institute of Urban Meteorology, China Meteorological Administration, Beijing 100089, China; mchen@ium.cn (M.C.); syfan@ium.cn (S.F.); jhe@ium.cn (J.H.); yjdou@ium.cn (Y.D.)
2 State Key Laboratory of Remote Sensing Science, Institute of Remote Sensing and Digital Earth, Chinese Academy of Sciences, Beijing 100101, China; shijc@radi.ac.cn
3 Joint Center for Global Change Studies (JCGCS), Beijing 100875, China
\* Correspondence: yhxie@ium.cn; Tel.: +86-10-6840-0741

**Abstract:** The Advanced Technology Microwave Sounder (ATMS) mounted on the Suomi National Polar-Orbiting Partnership (NPP) satellite can provide both temperature and humidity information for a weather prediction model. Based on the rapid-refresh multi-scale analysis and prediction system—short-term (RMAPS-ST), we investigated the impact of ATMS radiance data assimilation on strong rainfall forecasts. Two groups of experiments were conducted to forecast heavy precipitation over North China between 18 July and 20 July 2016. The initial conditions and forecast results from the two groups of experiments have been compared and evaluated against observations. In comparison with the first group of experiments that only assimilated conventional observations, some added value can be obtained for the initial conditions of temperature, humidity, and wind fields after assimilating ATMS radiance observations in the system. For the forecast results with the assimilation of ATMS radiances, the score skills of quantitative forecast rainfall have been improved when verified against the observed rainfall. The Heidke skill score (HSS) skills of 6-h accumulated precipitation in the 24-h forecasts were overall increased, more prominently so for the heavy rainfall above 25 mm in the 0–6 h of forecasts. Assimilating ATMS radiance data reduced the false alarm ratio of quantitative precipitation forecasting in the 0–12 h of the forecast range and thus improved the threat scores for the heavy rainfall storm. Furthermore, the assimilation of ATMS radiances improved the spatial distribution of hourly rainfall forecast with observations compared with that of the first group of experiments, and the mean absolute error was reduced in the 10-h lead time of forecasts. The inclusion of ATMS radiances provided more information for the vertical structure of features in the temperature and moisture profiles, which had an indirect positive impact on the forecasts of the heavy rainfall in the RMAPS-ST system. However, the deviation in the location of the heavy rainfall center requires future work.

**Keywords:** Advanced Technology Microwave Sounder (ATMS); radiance assimilation; heavy rainfall; rapid-refresh multi-scale analysis and prediction system—short-term (RMAPS-ST)

---

## 1. Introduction

Satellite observations have been playing an increasingly important role in numerical weather prediction (NWP) systems [1]. Information about atmospheric temperature and humidity structures and surface characteristics can be obtained from the measurements of satellite instruments through retrieval products [2,3]. This information, as a representative of truth, can be combined with short-range forecasts to produce more optimal initial conditions for the NWP system [4], which names data assimilation indirectly. In recent decades, the approach of directly assimilating satellite radiances has

been available because of the development of forward radiative transfer models. Radiative Transfer for Television and Infrared Observation Satellite Operational Vertical Sounder (RTTOV) and the Community Radiative Transfer Model (CRTM) have been widely employed as forward operators in data assimilation systems [5–7], which made full use of satellite radiances and significantly contributed to the improvement of forecast skills in NWP systems [8–11].

The direct assimilation of satellite radiances has become practical in the field of weather prediction since the 1990s [6], and the relative researches have also been active. As the first version of the RTTOV model was developed in 1992, satellite radiance data began to be routinely assimilated into the operational systems at the European Centers (EC) of Medium-Range Weather Forecasts (ECMWF) [8,12]. The cloud-cleared satellite radiance data from various instruments were incorporated into the operational systems at the National Centers for Environmental Prediction (NCEP) in 1995 [13]. More recently, the advancement in observation and assimilation techniques allowed for more applications from various new satellite instruments in weather prediction environments. These satellite observations provided above 90% of the actively assimilated data for weather prediction models and greatly improved the accuracy of NWP models [14,15]. Kanamitsu et al. [16] performed several 1-month experiments using the NCEP reanalysis system and pointed out that the impact of satellite data was significant for atmospheric analysis, especially over ocean areas with sparse data and in the stratosphere. Ye et al. [17] assessed the forecast results over a long-term sample from the NCEP global forecast system with satellite data and revealed that the reduced error of model forecasts had a remarkably high correlation with satellite observations. McNally [18] studied the assimilation of overcast infrared radiances in the system of ECMWF four-dimensional variation and obtained a small but statistically significant improvement in forecast quality. Joo et al. [19] compared the impacts of various observations using the method of adjoint-based forecast sensitivity to observations and found that satellite observations contributed about 64% in reducing the short-range forecast errors within the Met Office global NWP system. Lin et al. [20] reported that more added new satellite radiance observations can give a positive impact on the operational Rapid Refresh (RAP) system of the National Oceanic and Atmospheric Administration (NOAA).

The accuracy of precipitation forecasts produced by a numerical model is greatly influenced by the initial states of the atmosphere, especially for the short-term forecast system [2]. Satellite measurements can provide features of atmosphere humidity and temperature in the vertical direction with excellent coverage. Great efforts have been made to improve the accuracy of precipitation forecasts with various satellite observations in NWP models. Kazumori [11] demonstrated a considerable improvement of precipitation forecasts in the mesoscale system due to satellite radiance data assimilation, although the improvement was relatively small over land, with a deep convective area and limited-to-weak precipitation forecast over the ocean. Xu et al. [21] examined the influence of satellite radiance observations on the accuracy of forecasting precipitation in Southwest Asia. It was found that an improvement of the initial conditions can be obtained with satellite radiances in the NWP system, leading to a reduction of the forecast errors for most locations within the 24-h forecast range. Zou et al. [22] assimilated satellite radiances of microwave humidity sounding to forecast precipitation events with a new cloud detection algorithm and increased the threat scores of 3-h accumulated rainfall by about 50% after 3–6 h of the forecast range. Sagita et al. [23] investigated the impact of satellite radiances from several instruments on rainfall prediction in the Java region and pointed out that a higher accuracy of precipitation forecast could be obtained with radiance data assimilation in the system, even though the contribution from satellite observations was small. Wang et al. [24] suggested that the assimilation of water vapor radiances from the Advanced Himawari Imger (AHI) of Himawari-8 improved the initial wind and humidity fields in the forecast of heavy rainfall, which contributed to the accuracy of rainstorm prediction in the first 3–6 h.

The satellite of the Suomi National Polar-Orbiting Partnership (NPP), launched in 2011, was the first mission of the new-generation Joint Polar Satellite System (JPSS), developed by the National Aeronautics and Space Administration (NASA). The Advanced Technology Microwave Sounder

(ATMS) is a microwave instrument loaded on the Suomi NPP. It can provide a wide range of measurements about the surface and atmospheric characteristics for weather forecasting. The ATMS contains most temperature and humidity channels from the instruments of both the Advanced Microwave Sounding Unit-A (AMSU-A) and the Microwave Humidity Sounder (MHS), which have significantly contributed to current forecasting skills [25,26]. It also has added three new sounding channels. The newly added channels of ATMS can obtain detailed information about low tropospheric thermal and water vapor in vertical structures [6]. A main feature of ATMS is that it can measure temperature and moisture information of the same overlapping scene with different resolutions. This successful exploitation makes it more convenient to improve the quality control (QC) in the application of ATMS data assimilation [27]. Therefore, observations from the ATMS enable us to understand and improve the current weather forecasting skills. Bormann et al. [28] performed the initial assimilation trials with ATMS data at ECMWF and found that the analysis and forecast impact were significantly positive in the short range over the Southern Hemisphere and in the long range over the Northern Hemisphere. Weston et al. [29] enhanced the assimilation of ATMS radiances at ECMWF by accounting for inter-channel error correlations and tuning the error variances and showed that significant improvements to the first guess fits with independent observations. Xue et al. [30] assessed the impact of assimilating ATMS and Cross-track Infrared Sounder (CrIS) data on precipitation prediction over the Tibetan Plateau and obtained an improved precipitation pattern.

The aim of this study is to analyze and evaluate the impact of assimilating ATMS radiance observations on the forecast accuracy of heavy rainfall in the operational rapid-refresh multi-scale analysis and prediction system—short-term (RMAPS-ST). The rainstorm that occurred from 18 July to 20 July 2016, was investigated. It was one of the heaviest rainstorms in the past 60 years and caused widespread floods and major damage over North China [24]. However, the main features of this heavy rainfall event, such as the location and intensity, were not captured well in the RMAP-ST. Figure 1b shows the spatial pattern of 24 h of accumulated rainfall over North China on 19 July 2016, indicating that the heavy rainfall above 100 mm appeared mainly in Beijing and its southwest. The outline of this article is as follows: The introduction is provided in Section 1. The RMAPS-ST system, the characteristics of satellite radiance data, and their application methods are briefly introduced in Section 2. The experimental setting and QC of satellite radiances are described in Section 3. Section 4 first presents an analysis of brightness temperatures from ATMS observations against simulations from the forecasts of the RMAPS-ST based on departure statistics and then evaluates the results from two groups of experiments with and without assimilating ATMS radiances. Sections 5 and 6 give the results and conclusions, respectively.

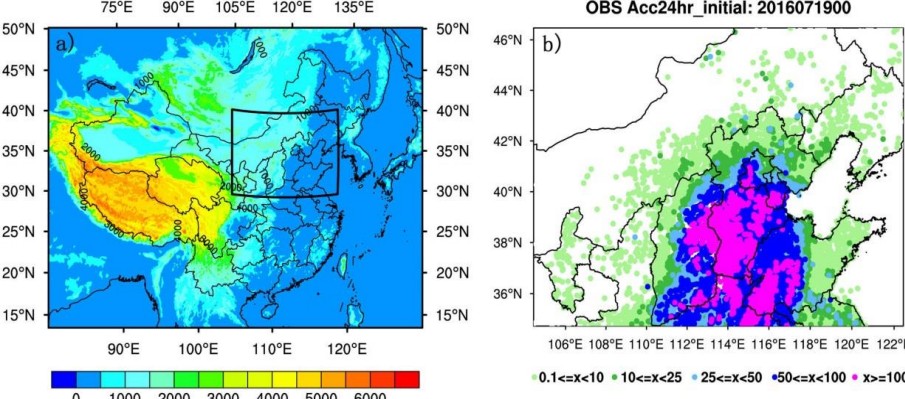

**Figure 1.** (**a**) Two nested domains in the rapid-refresh multi-scale analysis and prediction system—short-term (RMAPS-ST) with terrain elevation (m). The outside domain covered the China region and the nested domain, mainly located in North China (the inner black box). (**b**) Observations of 24-h accumulated rainfall on 19 July 2016, from ground stations.

## 2. Materials and Methods

### 2.1. The RMAP-ST System

The RMAPS-ST is a regional assimilation and forecast operational system at the Beijing Meteorological Bureau. It employs the Weather Research and Forecasting (WRFV3.8.1) model and its data assimilation (WRFDA) system [9,31,32]. Multi-source data assimilation, which has been developed by the Institute of Urban Meteorology, is one of the main features of the RMAP-ST system. The RMAPS-ST v1.0 was released in May 2017. It provides high-resolution gridded reanalysis data for both operational forecasts and relevant services. The RMAPS-ST mainly focuses on 0–12 or 0–24 h short-term forecasts.

The RMAPS-ST configuration is region-specific as it is used operationally over China. Figure 1a shows the two nested domains in the RMAP-ST system. The outer domain (Domain 1) mainly covers all of China with a relatively complex terrain in the West region. It has 649 × 500 grid points with a resolution of 9 km. The nested domain (Domain 2), which is located in North China, as the black box indicates in Figure 1a, has 550 × 424 grid points with a horizontal resolution of 3 km. A set of specially processed static data, such as land use, topography, and recently updated vegetation fraction with higher precision, were used in the system. In the vertical structure, 50 sigma levels were configured for both domains. The top pressure of the model was set to 50 hPa. Forecast products (0.25° × 0.25°) from the ECMWF were taken to provide the initial and boundary conditions every 3 h for the RMAPS-ST. The main physics parameterization employed in the RMAPS-ST included Thompson double moment microphysics, Kain–Fritsch deep convection, ACM2 PBL (Asymmetric Convective Model with non-local upward mixing and local downward mixing), and RRTMG (Rapid Radiative Transfer Model for GCMs) radiation schemes. The cumulus scheme was not activated in the inner domain. For data assimilation, the three-dimensional variation (3D-Var) method was employed in the RMAP-ST system. The background error used for data assimilation in the 3D-Var method was obtained using the domain-specific forecast error statistics of the system. It was generated with the model perturbations of over 1-month of forecasts through the method of the National Meteorological Center (NMC) [33]. Different types of conventional observations were incorporated into the system to improve the analysis for both domains in 3-h cycling runs. Most of these conventional data were collected every hour, mainly including meteorological terminal aviation routine weather report (METAR), aircraft meteorological data relay (AMDAR), synoptic (SYNOP), global positioning system derived zenith total delay (GPSZTD), oceanographic buoys (BUOY), and ship-based (SHIP) observations. Particularly, pilot balloon system (PILOT) or radiosonde observations (RAOB) were obtained twice a day, at 0000 and 1200 UTC (Coordinated Universal Time). Besides conventional observations, radar data assimilation, including radial velocity and reflectivity, was also performed in Domain 2 of the system.

### 2.2. ATMS Radiance Observations

Radiance observations from the instrument of ATMS were used in this study. The ATMS is a microwave radiometer with 22 measurement channels. It combines the heritage channels from AMSU-A and MHS, which are the predecessors of the ATMS. Three new additional channels include one temperature channel and two humidity channels. ATMS sounding channels 1–16 are mainly used to measure atmospheric temperature and channels 17–22 for atmospheric humidity. They have a cross-track swath width of 2300 km with a spatial sampling of 1.11° at 96 scan positions. Table 1 gives the central frequency of ATMS channels and their weighting function peaks [27].

Window channels 1 and 2 of the ATMS are commonly employed for QC in the system of data assimilation. They have a much larger field of view (FOV) size (75 km at nadir) than the other sounding channels (32 or 16 km at nadir). Temperature-sounding channels 3–16 and humidity-sounding channels 17–22 have FOVs with 32 km and 16 km at nadir, respectively. The spatial sampling of these channels is denser with a smaller footprint but brings larger noise compared with that of the AMSU-A [28]. To reduce the noise and achieve a performance comparable to AMSU-A for data assimilation in NWP, the

averaging of ATMS footprints should be performed. There are several averaging approaches developed by the European Organization for the Exploitation of Meteorological Satellites (EUMETSAT) used for ATMS data, including Backus–Gilbert weighted averaging, neighboring three scan-positions and scan-lines (3 × 3 averaging) averaging, and Fourier-based methods. In our experiments, the fast Fourier transformation (FFT) technique was applied to ATMS channels 3–22 to make the spatial average.

**Table 1.** Central frequency of each channel and its weighting function (WF) peak for the Advanced Technology Microwave Sounder (ATMS).

| Channel Number | Frequency (GHz) | WF Peak (hPa) | Channel Number | Frequency (GHz) | WF Peak (hPa) |
|---|---|---|---|---|---|
| 1 | 23.8 | Surface | 12 | 57.2903 | 25 |
| 2 | 31.4 | Surface | 13 | 57.2903 ± 0.322 | 10 |
| 3 | 50.3 | Surface | 14 | 57.2903 ± 0.322 ± 0.010 | 5 |
| 4 | 51.76 | 950 | 15 | 57.2903 ± 0.322 ± 0.004 | 2 |
| 5 | 52.8 | 850 | 16 | 88.20 | Surface |
| 6 | 53.596 ± 0.115 | 700 | 17 | 165.5 | Surface |
| 7 | 54.4 | 400 | 18 | 183.31 ± 7 | 800 |
| 8 | 54.94 | 250 | 19 | 183.31 ± 4.5 | 700 |
| 9 | 55.5 | 200 | 20 | 183.31 ± 3 | 500 |
| 10 | 57.2903 | 100 | 21 | 183.31 ± 1.8 | 400 |
| 11 | 57.2903 ± 0.115 | 50 | 22 | 183.31 ± 1.0 | 300 |

*2.3. Verification Strategy*

In this study, observations from conventional sounding and ground-based stations were used for comparison and evaluation. A series of quality control methods have been performed for these observations in advance [34,35]. There are more than 100 sounding sites over Domain 1 of the RMAPS-ST, as shown in Figure 2. Conventional sounding can provide atmospheric temperature, humidity, and wind observations every 12 h per day, at 0000 UTC and 1200 UTC. Ground automatic weather stations (AWSs) mainly provide near-surface humidity, temperature, and precipitation observations with high temporal frequency. Rain gauges from AWSs are used to obtain rainfall rates with reliable accuracy [36]. For comparison and evaluation, it is necessary to match these observations and forecast gridded model values in horizontal and vertical directions. In the horizontal direction, the gridded point values from the forecast were assigned to the nearest ground observations through the nearest neighbor method. In the vertical direction, the interpolation method was used where forecast and observation were at different vertical levels. The forecast gridded point value was then interpolated to the level of the corresponding observation with the natural logarithm of pressure coordinate. If the vertical levels of forecast and observation were the same, they were directly matched.

To quantitatively assess the impact of assimilating the ATMS radiance data on the forecasts of heavy rainfall that occurred over North China in the RMAPS-ST, several score skills were used for the evaluation of quantitative precipitation forecasts against observations. These statistical indices include the bias score (BIAS), false alarm rate (FAR), and Heidke skill score (HSS) [37–39]. They were all calculated on the basis of a joint-distribution contingency table (Table 2), which is a useful way to describe the forecast performance against observations. There are four combinations in the contingency table, *hits*, *misses*, *false alarms*, and *correct rejects*. Given a threshold, yes or no statement indicates whether the precipitation is equal or greater than the threshold. Then, *hit* means that the rainfall values from both forecast and observation equaled or exceeded the given threshold; *false alarm* means that the value of forecast was at or above the threshold and the corresponding observation was below the threshold; *miss* indicates that the value of the forecast was below but the observation was equal or greater than the threshold; *correct rejects* means that the values from both forecasts and observations

were below the given threshold. Thus, the indices of BIAS, FAR, and HSS can be expressed through four components, *a*, *b*, *c*, and *d*, as the following:

$$BIAS = \frac{a+b}{a+c} \tag{1}$$

$$FAR = \frac{b}{a+b} \tag{2}$$

$$HSS = \frac{(a+d) - (expected\_correct)_{random}}{N - (expected\_correct)_{random}} \tag{3}$$

where $N = a + b + c + d$,

$$(expected\_correct)_{random} = \frac{1}{N}[(a+c)(a+b) + (d+c)(d+b)] \tag{4}$$

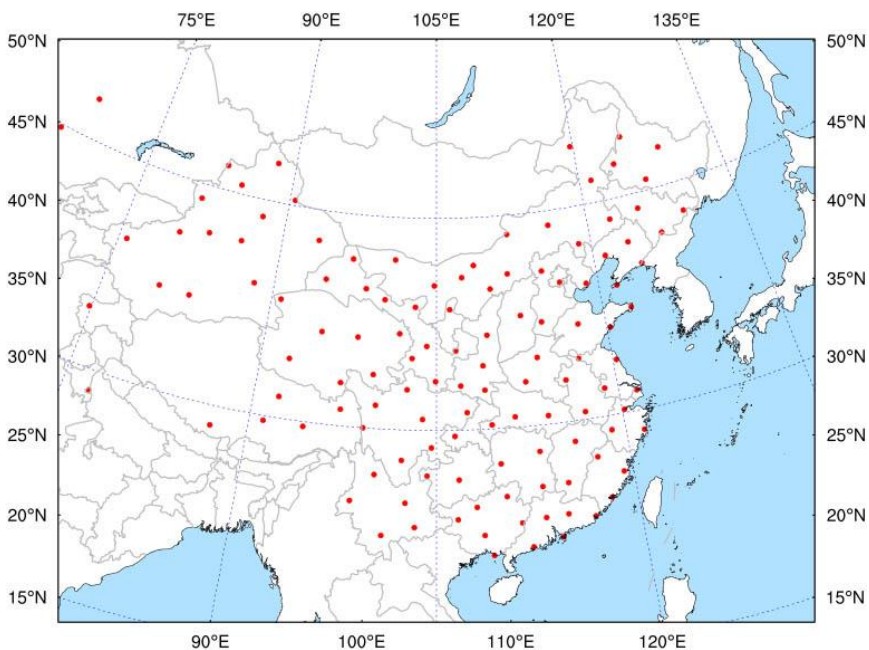

**Figure 2.** Conventional sounding observations in Domain 1 of the RMAPS-ST used for verification and evaluation in the data assimilation experiments.

**Table 2.** Contingency table.

|  |  | Observed | |
|---|---|---|---|
|  |  | **yes** | **no** |
| **Forecast** | **yes** | *hits* (*a*) | *false alarms* (*b*) |
|  | **no** | *misses* (*c*) | *correct rejects* (*d*) |

The BIAS ranged from 0 to infinity. The value of 1.0 represented an unbiased forecast. The BIAS below 1.0 indicated a tendency of under-prediction (BIAS < 1) for the forecast system and over-prediction with a score above 1.0 (BIAS > 1). The FAR also had a range of 0 to 1, and 0 was the perfect score. It indicated the ratio of the forecast rain events actually did not occur. The HSS measured the fractional improvement of the forecasts after eliminating those correct forecasts purely by chance. It ranged from minus infinity to 1. Negative values mean that the chance forecast is better. 0 is no skill and 1 indicates a perfect forecast.

Additionally, the spatial characteristics of rainfall distribution from forecasts was evaluated against observations through statistic metrics of mean error (ME), mean absolute error (MAE), and Nash–Sutcliffe (NS) efficiency coefficient [39,40]. They are defined as:

$$ME = \frac{1}{N} \sum_{i=1}^{N} (F_i - O_i) \tag{5}$$

$$MAE = \frac{1}{N} \sum_{i=1}^{N} |F_i - O_i| \tag{6}$$

$$NS = 1 - \frac{\sum_{i=1}^{N} (O_i - F_i)^2}{\sum_{i=1}^{N} \left(O_i - \overline{O}\right)^2} \tag{7}$$

where $F_i$ represents the value in the $i$th forecast gridded point and $O_i$ is the corresponding observations. $\overline{O}$ is the mean value of the $O_i$ fields over the whole domain. The NS efficiency is a normalized statistic; 1 corresponds to a perfect forecast, and 0 indicates that the forecasts are as accurate as the mean of the observations. A minus value indicates that the forecast is no better than using the observed mean.

## 3. Assimilation Experiments

### 3.1. Experiment Setup

On the basis of the RMAPS-ST, two parallel assimilation experiments were carried out to produce the forecasts for the heavy rainfall that occurred from 18 July to 20 July 2016, over North China. In the first control experiment (CTRL), various conventional observation data were assimilated in the main domain (Domain 1), which would provide boundary conditions for the inner domain (Domain 2). For Domain 2, centered in North China, both conventional observations and radar data, including radial velocity and reflectivity, were used in the system but assimilated in two steps. That is, conventional data were first assimilated in the system to get an analysis and then radar data were assimilated based on the result from the first step to obtain the final analysis. The time window of assimilation was set to ±1.5 h from the analysis time. Conventional observations were collected every hour and radar data were collected every 6 min. The two types of observations were assimilated every 3 h to provide an improved analysis for each forecast in the system. In the second experiment of radiance data assimilation (DA_RAD), the configuration was the same as that of the CTRL, except for the addition of assimilating ATMS radiance data for Domain 1. For DA_RAD, the forward operator of the CRTM developed by the U.S. Joint Center for Satellite Data Assimilation (JCSDA) was used to calculate the top radiance from the model initial fields, corresponding to the satellite measurement [41]. The 3D-Var technique was used to perform the data assimilation experiments.

Both CTRL and DA_RAD experiments were conducted for 3 days from 18 July to 20 July 2016, based on the RMAPS-ST. For each day, a spin-up of 6-h run was first conducted from 1800 UTC of the day before. Additionally, 24-h forecasts, starting at 0000 UTC the next day, were then produced in 3-h cycling runs, as used operationally, as shown in Figure 3, and a total of 24 forecasts were yielded for each group of experiments.

### 3.2. Quality Control

Monitoring the observation quality is one of the most important aspects of ATMS data assimilation. QC and detection for ATMS observations should be performed strictly before they are taken into the data assimilation system. Several procedures of QC that have been widely used for satellite radiance observations are available for ATMS radiance data in the system. Channel selection, limb removal, cloud detection, bias correction (BC), and data thinning were the main QC procedures for ATMS data

in this study. Meanwhile, the assigned observation error for each channel provided by WRFDA was used in the ATMS data assimilation experiments.

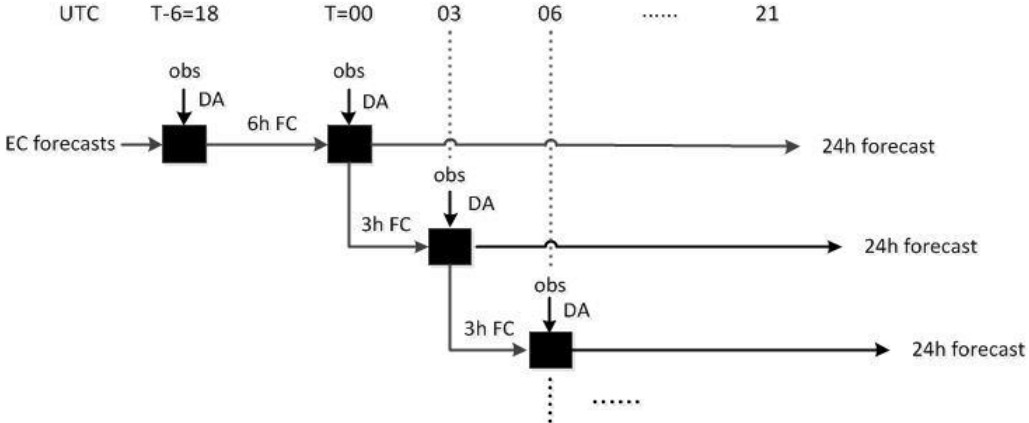

**Figure 3.** The schematic of data assimilation (DA) and forecast (FC) experiments in the RMAPS-ST.

Due to large errors and uncertainties, window channels with the peak of weight function near the surface were not used for assimilation. Therefore, ATMS channels 1–5 and 16–17, with some surface sensitivity, were first rejected over the land. Meanwhile, channels that had the peak of their weight function above the vertical limit of the system, 50 hPa, were also not considered for assimilation, including channels 11–15. Thus, the radiance data only from the ATMS sounding channels 6–10 and 18–22 were used in the assimilation system.

Further quality checks were performed for all the remaining sounding channels. The sounding instrument scans away from nadir had a longer atmospheric path with an associated increase in absorption [42], which can result in a reduction in the observed radiance. Limb observations were first removed when the zenith angle was larger than 45° for all channels. If the absolute value of the first guess departure was greater than 15 K or beyond three times the observation error standard deviation, observations were not used for data assimilation. For ATMS channels 6–8 and 18–22, observations contaminated by cloud or precipitation were discarded if the cloud liquid water was greater than the given threshold of 0.2 kg/m$^2$ [43]. Observations were also rejected when the absolute value of departure from channel 3 was higher than 5 K. For humidity-sounding channels 18–22, observations were not used for assimilation if the corresponding difference (referred as the scatter index) between channels 16 and 17 was larger than 3 K. Additionally, systematic bias should be treated correctly for ATMS radiance data before they are assimilated into the system. The scheme of bias predictors developed by Harris and Kelly [44] was used for bias correction. In this scheme, the bias of ATMS radiance data can be expressed through a set of predictors, as:

$$e = \sum_{i=1}^{n} \beta_i p_i \qquad (8)$$

where $p_i$ is the $i$th predictor and $\beta_i$ is the corresponding coefficient. The predictors used in our assimilation experiments included the scan position, the surface skin temperature, the total column water vapor, and the layer thicknesses of 300–1000 hPa and 50–200 hPa.

A thinning of 120 km was performed for all ATMS radiance data to reduce data redundancy. Figure 4 gives the distributions of brightness temperatures from ATMS channel 9 in the 9-km domain of the RMAPS-ST before and after QC at 1800 UTC 18 July 2016. It can be seen that the observations retained were small through all QC procedures.

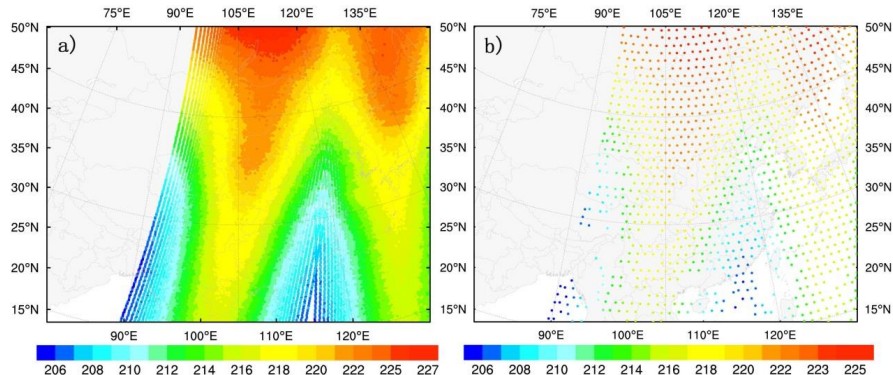

**Figure 4.** Distributions of brightness temperatures (K) calculated from ATMS radiances of channel 9 (**a**) before and (**b**) after quality control at 1800 UTC 18 July 2016, in the main domain of the RMAPS-ST.

## 4. Results

In this section, we first analyze the statistical features of departures against measured brightness temperatures from ATMS radiances. Then, the results of our assimilation experiments with and without ATMS radiance data are described on the basis of the RMAPS-ST.

### 4.1. Analysis of Departure Statistics

The instrument of ATMS can provide observations for the RMAPS-ST twice a day at around 0006 UTC and 1800 UTC. Figure 5 shows the numbers and statistics of the relative background (B) and analysis (A) departures against observations (O) for ATMS sounding different channels used for assimilation in Domain 1 valid at 1800 UTC 18 July 2016. Because of the influence of cloud or rain, the observation numbers of humidity channels 18–22 that were actually used for assimilation were less than those of temperature channels 6–10 after all QC procedures. For temperature channels 6–10, the standard deviation (Stdv) and the mean bias of the relative background departures after bias correction (REL_OMB_wb) were smaller than the values prior to correction (REL_OMB_nb), especially for channels 9–10. Here, REL_OMB_wb and REL_OMB_nb were referring to (O-B)/O with and without bias correction, respectively. The bias of the relative analysis departures (REL_OMA), referring to (O-A)/O, reduced further for each channel. Similarly, bias correction also reduced the standard deviation and the mean bias of the relative background departures for humidity channels 18–22. However, the errors of humidity channels were still higher than the values of temperature channels for the relative background and analysis departures. In comparison with the relative background departures, the standard deviations of the relative analysis departures were further reduced for humidity-sounding channels, which were much smaller and closer to 0.

Figure 6 gives the scatter plots of brightness temperatures (K) simulated by CRTM against observations from ATMS channels 9 and 20 at 1800 UTC 18 July 2016. For temperature-sounding channel 9, the brightness temperatures calculated from the background (or first guess) gave a good agreement with observations from ATMS data based on the statistics of 1097 points. The root-mean-square error of the background decreased from 0.591 to 0.351 K through bias correction. The corresponding values of the analysis were further decreased, from 0.351 to 0.185 K for the root-mean-square error and from 0.341 to 0.184 K for the standard deviation, respectively. For the humidity-sounding channel 20, the amount of radiance data used for assimilation was much less, with 486 points. After bias correction (with BC), the root-mean-square error of the background was still relatively large, only reduced from 2.084 to 1.8 K. The corresponding values of the analysis were greatly reduced from 1.833 to 0.590 K (by about 67.8%) for the root-mean-square error and from 1.820 to 0.568 K (by about 68.8%) for the standard deviation, respectively. These results indicate that bias correction can reduce the systematic error to some extent and the analysis with the assimilation of ATMS radiances was closer to the observations than the first guess.

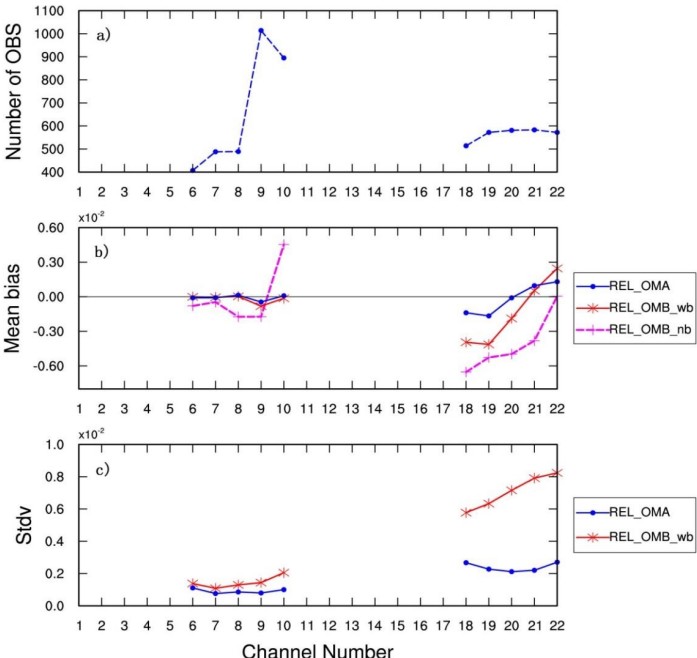

**Figure 5.** The observation numbers (**a**) used in data assimilation and the statistics of the relative departures for ATMS different channels. (**b**) Mean bias and (**c**) standard deviation of the relative background departures with (REL_OMB_wb) and without (REL_OMB_nb) bias correction, and the relative analysis departures (REL_OMA) against the observed brightness temperatures at 1800 UTC 18 July 2016.

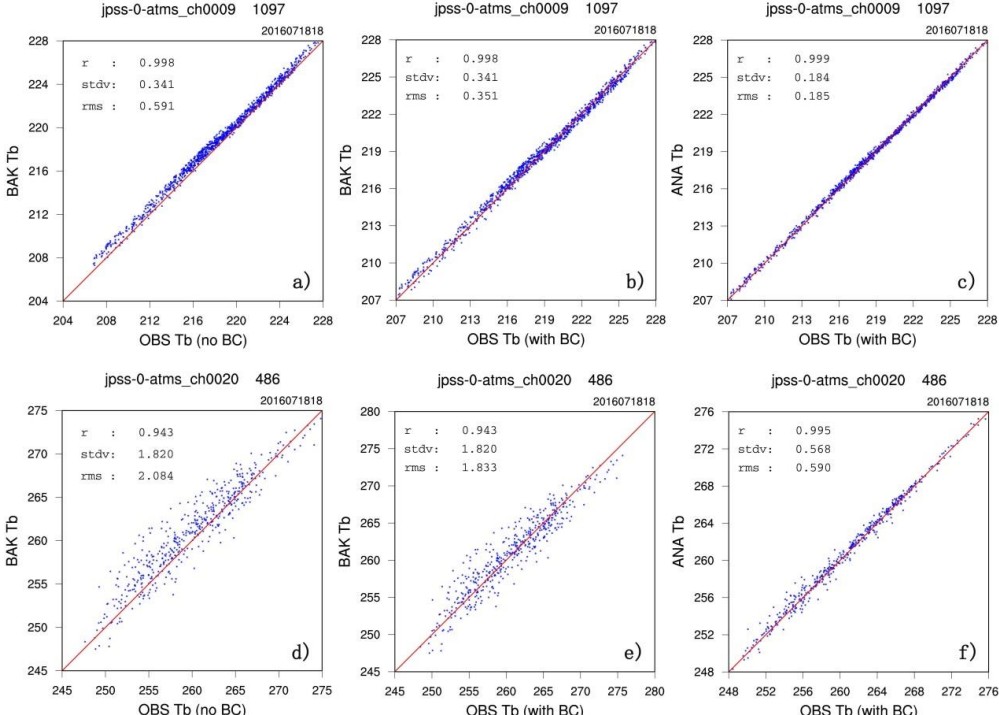

**Figure 6.** The scatter plots of brightness temperatures (K) simulated by CRTM against observations (O) from ATMS channels 9 and 20 at 1800 UTC 18 July 2016. (**a**,**d**) and (**b**,**e**): Background (BAK) versus observations before (no BC) and after (with BC) bias correction, respectively. (**c**,**f**): Analysis (ANA) versus observations.

*4.2. Verification and Evaluation*

The forecast results of temperature, humidity, and wind from two groups of experiments with and without ATMS radiance data assimilation, CTRL, and DA_RAD, were first compared against conventional radiosonde observations. Figure 7 shows the error statistics in vertical levels for temperature and humidity over Domain 1 at 0 h of forecasts. For temperature, the average bias in DA_RAD was reduced between 300 and 500 hPa and for the levels 100 and 850 hPa. However, the root-mean-squared-error (RMSE) did improved consistently. For humidity, the average bias of DA_RAD decreased at the levels 300 and 700 hPa in comparison with that of CTRL, whereas the RMSE of humidity in DA_RAD had an overall improvement for the vertical profile, except for the level 850 hPa. The assimilation of ATMS radiances also had an indirect impact on wind fields through multivariate correlations referring to the background error covariance matrix. In comparison with that of CTRL, the average bias of zonal wind (UGRD) had a relatively noticeable reduction at levels 300 and 700 hPa and a slight overall improvement can be found for the RMSE of the vertical profile, except for the level 500 hPa. For the meridional wind (VGRD), the experiment of DA_RAD had a little smaller RMSE than that of CTRL for the vertical profile, except for the levels 400 and 925 hPa. It can be seen that the assimilation of ATMS radiance data brought some added value for the initial fields, especially for humidity.

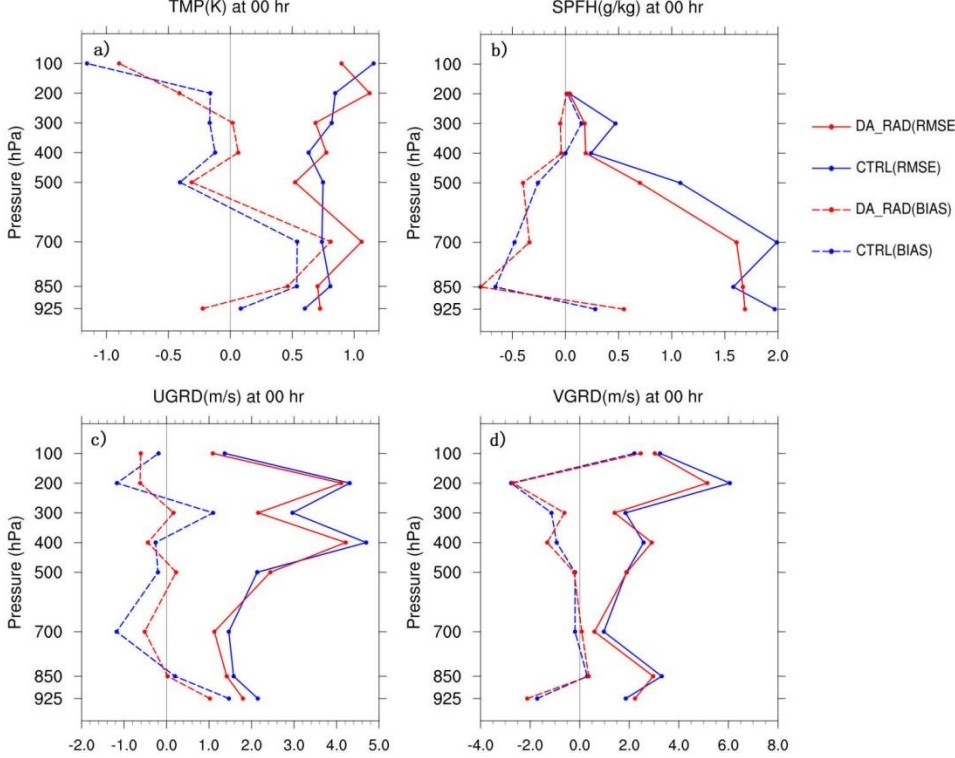

**Figure 7.** The error statistics in vertical levels for temperature, humidity, and wind at the time 0 h of forecasts against sounding observations, including the average bias and RMSE for (**a**) temperature (TMP), (**b**) specific humidity (SPFH), (**c**) zonal wind (UGRD), and (**d**) meridional wind (VGRD).

Figure 8 shows the average errors and RMSEs of forecasting wind at 10 m, temperature, and humidity at 2 m over the forecast range of 0–24 h. It can be seen that assimilating ATMS radiance data had little impact on the near-surface components. The average errors of humidity and temperature at 2 m in DA_RAD reduced slightly after the 6-h forecast compared with that in CTRL. Meanwhile, almost no difference for the corresponding RMSEs between the two experiments was obtained. For

the wind at 10 m, the RMSE even had a small increase after the 12-h forecast with the ATMS radiance data assimilation.

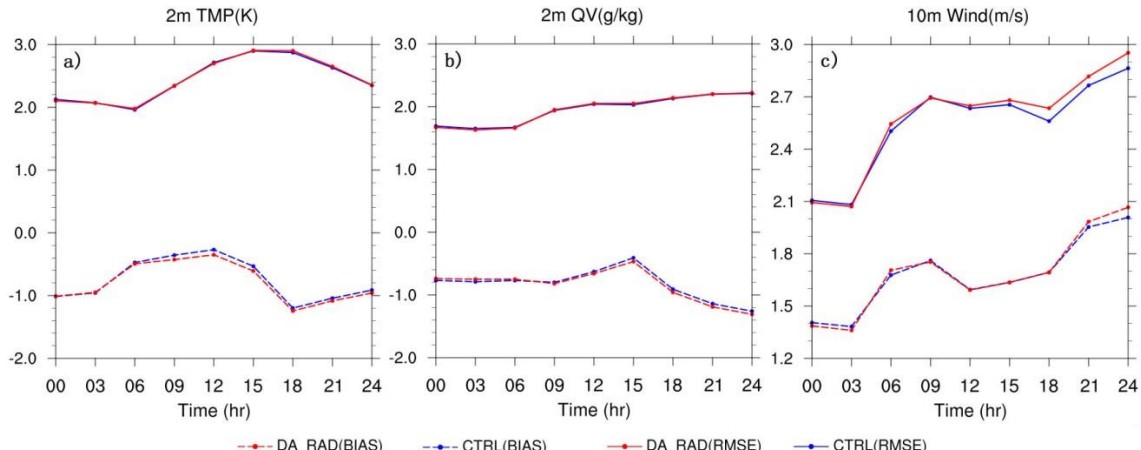

**Figure 8.** The average errors (bias score (BIAS)) and RMSEs of forecasting (**a**) temperature (TMP) at 2 m, (**b**) humidity (QV: QVAPOR) at 2 m, and (**c**) wind at 10 m over the forecast range of 0–24 h for CTRL and DA_RAD.

The rainfall forecasts over Domain 2 from experiments before (CTRL) and after (DA_RAD) assimilating ATMS radiance data were evaluated through score skills against rainfall observations from about 850 ground stations. Statistical indices HSS, BIAS, and FAR provided details for the score skills of 6-h accumulated rainfall from the two experiments, as shown in Figure 9. It can be seen that the HSS of DA_RAD was greater than that of CTRL almost for all thresholds over the 0–24 h forecast range. In the first 6-h lead time, the HSS scores of the DA_RAD had a noticeable increase for the thresholds above 25.0 mm, indicating a higher accuracy of heavy rainfall forecast relative to that of random chance. The BIAS values of rainfall forecasts in both experiments decreased with the rainfall threshold. Above the 5 mm rainfall, the BIAS values of DA_RAD were below 1 and smaller than those of CTRL, suggesting a more underestimated rainfall forecast against the corresponding site observations. However, the corresponding FARs of DA_RAD were also lower than those of CTRL, although they both increased with the rainfall threshold. It suggests that assimilating ATMS radiance data in the RMAP-ST system reduced the fraction of false alarms for rainfall forecasting in the first 6 h, especially for heavy rainfall above 50 mm. For the 6–12 h forecast range, the BIASs of CTRL were all above 1, which indicated an over-prediction of the rainfall forecasts. In comparison with the CTRL, the BIAS values were closer to 1 for all thresholds, except for 5 mm, which indicated an improvement for the over-prediction of the CTRL. The FAR values of DA_RAD were also smaller than those of CTRL. For the 12–18 h forecast range, the BIAS of DA_RAD was closer to 1 for rainfall above 25 mm, which presented an improvement for the under-prediction of the CTRL. In the forecast range of 18–24 h, a significant over-prediction was obtained in the DA_RAD for the rainfall above 50 mm. In addition, the difference of FAR values between the two experiments decreased during the last 12 h of the forecast range.

Figure 10 shows the statistic metrics of ME, MAE, and NS efficiency coefficient for forecasts against observed hourly rainfall averaged spatially over Domain 2. It can be seen that the ME values of the two experiments were both close to 0. In the first 8-h lead time, the minus values of DA_RAD indicated a lower forecasting rainfall compared with observations. The corresponding MAE values of DA_RAD were also lower than that of CTRL in the first 8 h of the forecast range. It suggested that the absolute errors of rainfall had been reduced early in the heavy rainfall forecast with the ATMS radiance data, although the average forecast rainfall was a little low. Similarly, the assimilation of the ATMS radiance data brought the NS coefficient closer to 0 for the first 10 h, indicating a better spatial agreement of DA_RAD forecasts with the average of observations compared with that of CTRL forecasts during this period.

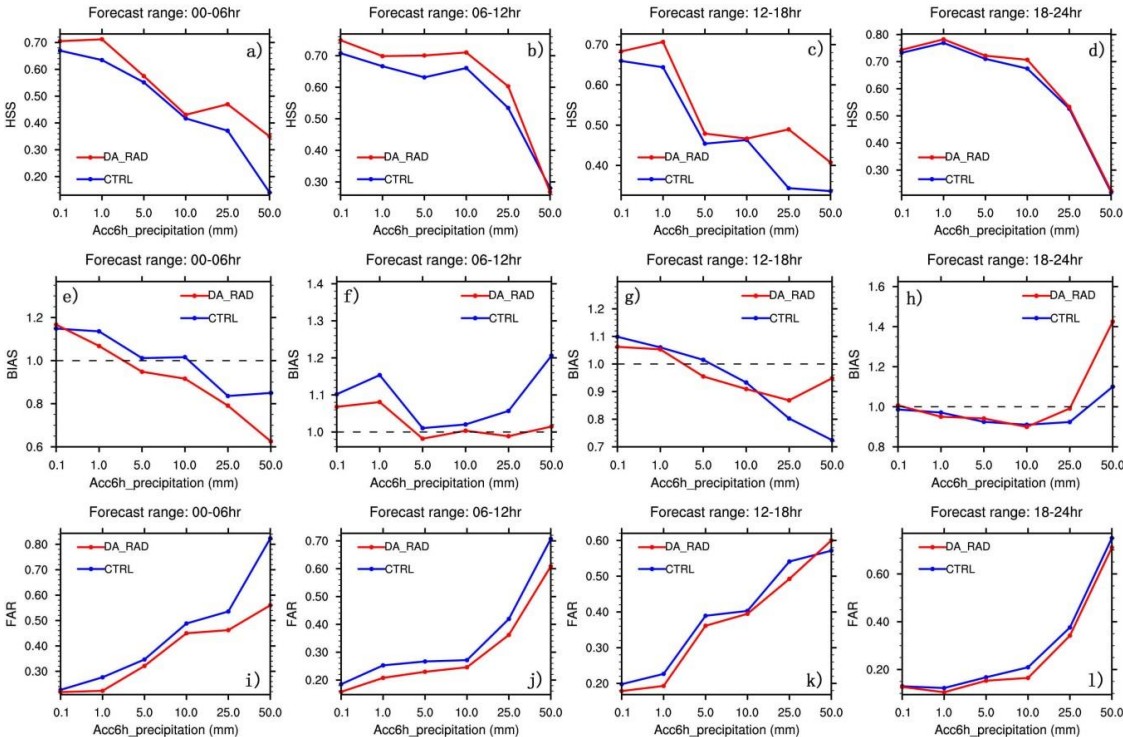

**Figure 9.** Statistical indices of the Heidke skill score (HSS) (**a**–**d**), bias score (BIAS) (**e**–**h**), and false alarm rate (FAR) (**i**–**l**) for 6-h accumulated rainfall from the forecasts before (CTRL) and after (DA_RAD) assimilating ATMS radiance data.

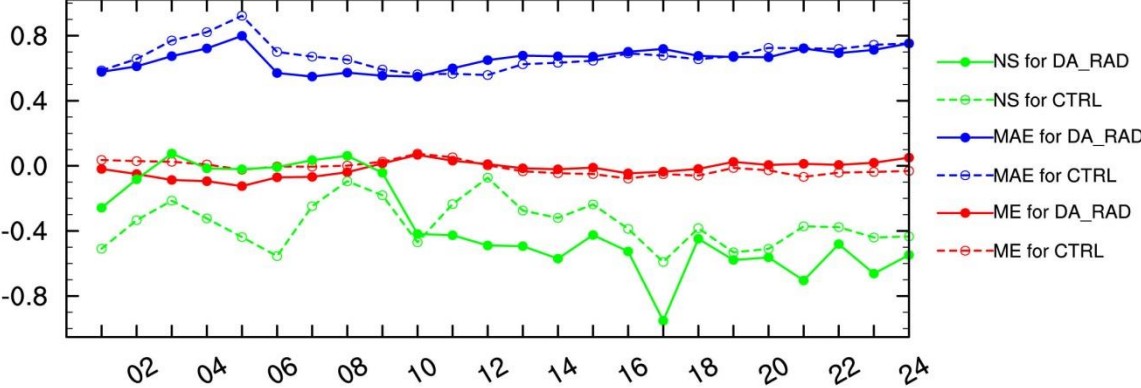

**Figure 10.** Statistic metrics of mean error (ME), mean absolute error (MAE), and Nash–Sutcliffe (NS) efficiency coefficient for forecasts against observed hourly rainfall over Domain 2 with a resolution of 3 km.

Figure 11 gives the spatial patterns of 6-h accumulated rainfall distribution in Domain 2 during 19 July 2016. Rainfall observations from ground stations that were used to evaluate score skills of rainfall forecasts are shown as the first line in Figure 11a–d. The corresponding forecast results from the two experiments CTRL and DA_RAD are shown in Figures 11e–h and 11i–l, respectively. At 0000 UTC, there was a noticeable over-prediction for the forecast rainfall in CTRL at locations indicated by the three red boxes, compared with observations. An improvement can be found in DA_RAD forecasts for the corresponding over-prediction and even an under-prediction appeared at the main center of the heavy rainfall in the southwest. At 0600 UTC, the spatial patterns of 6-h accumulated rainfall in DA_RAD agreed with the observations better than those in CTRL, especially for the location in the red box. At 1200 UTC and 1800 UTC, the red boxes indicated the main differences of heavy rainfall between the two experiments. Compared with that in CTRL, ATMS radiance data assimilation

increased the scope of heavy rainfall in DA_RAD, which gave a good agreement with observations in spatial distribution patterns. However, the verification and evaluation greatly depend on the number and spatial distribution of ground stations.

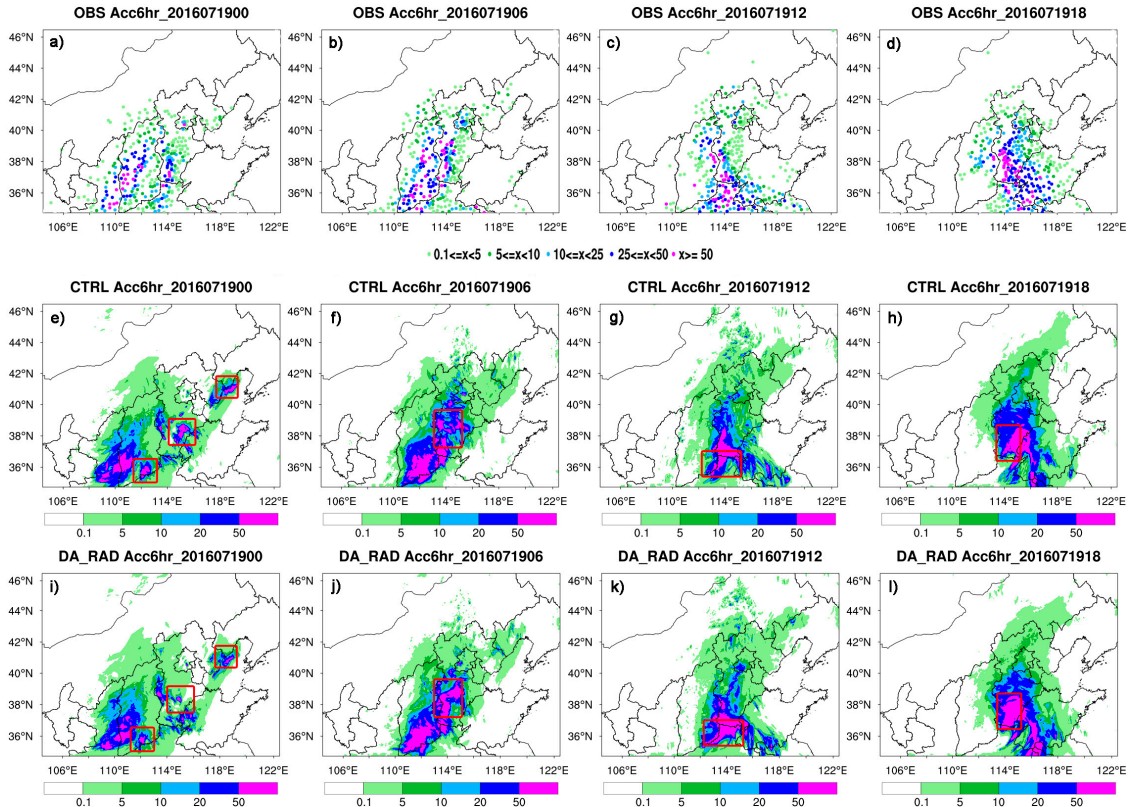

**Figure 11.** Spatial patterns of 6-h accumulated rainfall distribution from (**a–d**) observations based on ground stations, (**e–h**) forecasts of CTRL, and (**i–l**) forecasts of DA_RAD at 0000 UTC, 0600 UTC, 1200 UTC, and 1800 UTC during 19 July 2016, respectively.

## 5. Discussion

The assimilation of ATMS radiance data brought some added value for the initial conditions in the RMAPS-ST based on the 3D-Var technique, especially for humidity. In comparison with the CTRL experiments, assimilating ATMS radiance data reduced the average errors and RMSEs of initial moisture and temperature on the middle levels. It also indirectly improved the initial wind fields on the low levels to some extent through multivariate correlations referring to the background error covariance matrix. As a result, ATMS radiance data assimilation reduced the mean absolute errors within the lead time of the forecasts (0–10 h) and further improved the score skills of quantitative rainfall forecast in the RMAPS-ST system during heavy rainfall. However, errors of temperature and humidity in vertical levels were still large, especially for low levels. In addition, due to under- or over-prediction in some local areas, rainfall forecasts still did not agree well with observations, which resulted in overall low score skills for the two experiments.

First, ATMS radiances data only provide information on temperature and humidity, which have a limited effect on the atmospheric process and circulations. Observations from ATMS that actually used for data assimilation were mainly between 300 and 700 hPa. Therefore, the positive impacts of ATMS data on initial conditions were mainly reflected in these levels, especially for humidity. Because of the influence of terrain height and its uncertainties, errors of temperature and humidity below 700 hPa were relatively large in the system. Figure 12 gives the relative humidity cross sections along 38.5 °N of the first guess and the analysis at 1800 UTC 18 July 2016. A wider distribution was found

for the relative humidity above 90% of the analysis than that of the first guess. The relative humidity above 90% of the analysis was also strengthened between 300 and 700 hPa. Figure 13 shows the cross sections of the vertical velocity along 38.5 °N from the 12-h forecasts of the two experiments initialized at 1800 UTC 18 July 2016. A clear updraft was found in the middle levels between 114 °E and 115 °E, leading to a stronger rainfall forecast in DA_RAD. It suggested that ATMS radiance observations provided more beneficial information for the initial conditions and indirectly strengthened the local updraft near the center area of the heavy rainfall, which contributed to the water vapor transport in the vertical direction and consequently improved the heavy rainfall forecasts. Nevertheless, the scope of influence from the ATMS radiance data was quite limited, although it brought an obvious positive impact on the heavy rainfall forecast in the first 6 h lead time.

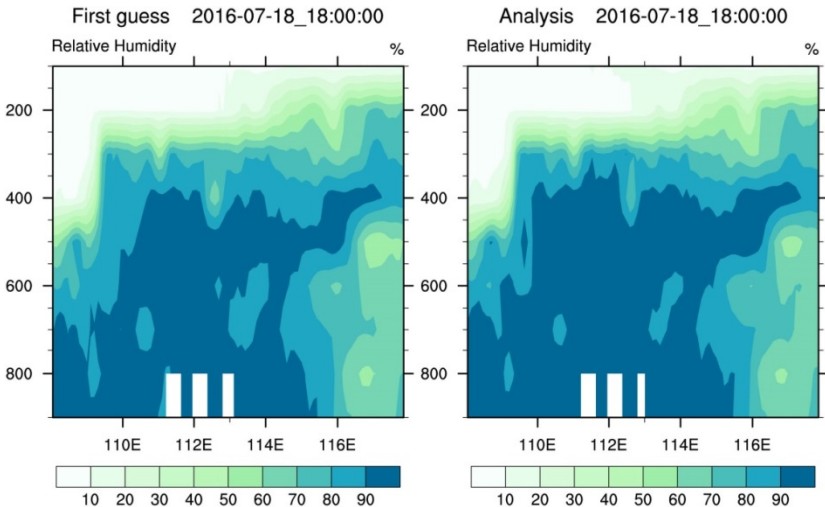

**Figure 12.** Cross sections of relative humidity (%) along 38.5 °N for the first guess and the analysis at 1800 UTC 18 July 2016.

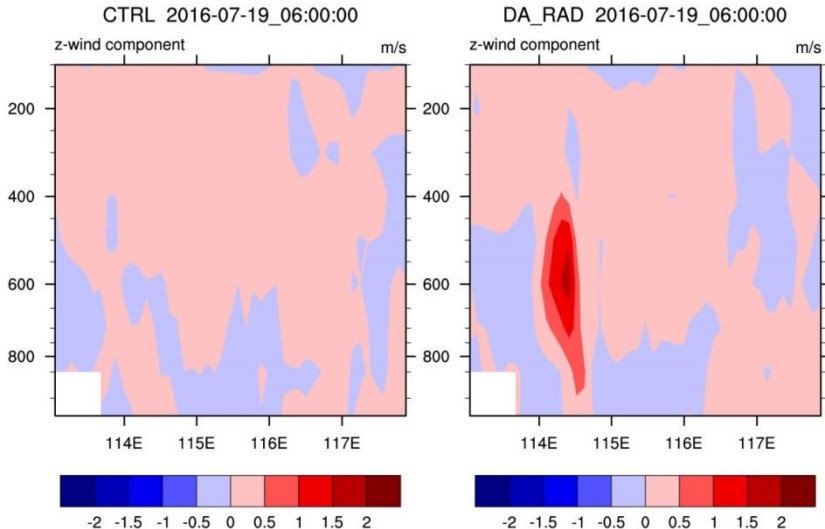

**Figure 13.** Cross sections of vertical velocity (m/s) along 38.5 °N from the 12-h forecasts of CTRL and DA_RAD experiments initialized at 1800 UTC 18 July 2016.

Figure 14 shows the geopotential height and wind fields from forecasts of the two experiments initialized at 1800 UTC 18 July 2016, for 500 and 700 hPa. The corresponding fields from EC were also used for comparison. For the level of 500 hPa, there was a clear low-pressure center located about 40 °N from the EC analysis fields. Additionally, a low-pressure center could be found in the CTRL

experiment, but it had a weak intensity and a southward position offset. In the experiment of DA_RAD, the assimilation of the ATMS radiance data strengthened the intensity of the low-pressure center and extended the widespread, but the location tended toward southward. For 700 hPa, the center of low pressure was located about 38.5 °N in the EC analysis fields, but the low-pressure system in the CTRL was obviously weak. In the DA_RAD, the assimilation of the ATMS radiance data ha little impact on the low-pressure system. That is, the main characteristics of the atmospheric processes and conditions still could not be captured well with the development of the atmospheric system, even though ATMS radiance observations were added.

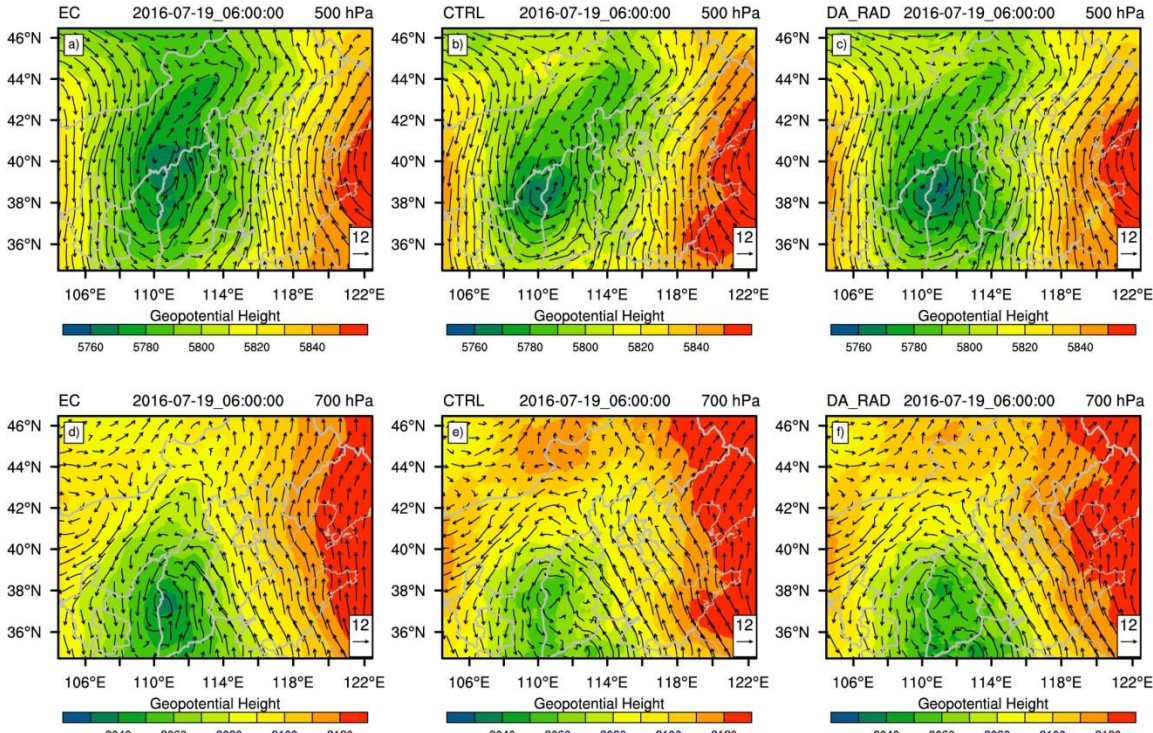

**Figure 14.** Geopotential height (shaded; m) and wind fields from the forecasts of CTRL and DA_RAD, initialized at 1800 UTC 18 July 2016, for (**a**–**c**) 500 hPa and (**d**–**f**) 700 hPa. The corresponding fields from (**a**,**d**) EC are used for comparison.

Second, the lower model top of 50 hPa directly affected the amount of ATMS data used for assimilation and the effects on upper levels. Early in the development of the RMAPS-ST system, conventional observations were the main source used for data assimilation, which considered 50 hPa as a suitable height. With the development of the system and sufficient computing resources, the top pressure of 10 hPa and 51 vertical levels will be adopted in the future version, which may facilitate the use of ATMS observations and more satellite data in the system.

Finally, observations and verified strategy are also the main factors influencing the evaluation of results. Both the quality and distribution of observations used for comparison are critical to decide score skills. In addition, the method of interpolation and matching between forecasts and observations also influence the errors.

## 6. Conclusions

New satellite measurements from ATMS can profile both atmospheric temperature and humidity fields and improve the quality of the NWP model. On the basis of the RMAPS-ST, this study evaluated the impact of assimilating ATMS radiance observations on the forecast of heavy rainfall that occurred from 18 July to 20 July 2016, over North China. Two groups of experiments with and without the assimilation of ATMS radiance data were conducted in 3-h cycling runs. Forecast results from the two

groups of experiments, including quantitative rainfall, temperature, moisture, and wind fields, were investigated and compared against observations.

It was found that ATMS radiances can provide beneficial information for the initial conditions in the RMAP-ST system. Errors of initial moisture between 300 and 700 hPa have a significant reduction after assimilating ATMS radiances. Consequently, the score skills of heavy rainfall forecasts have been improved with ATMS radiance data assimilation in the RMAPS-ST. Compared with the experiment assimilating only conventional observations, the assimilation of ATMS radiance data increased the HSS scores of 6-h accumulated precipitation almost for the short-range forecasts of 0–24 h, especially for heavy rainfall above 25 mm in the first 6-h lead time. However, errors of initial conditions were still large in the upper and lower levels and the forecast rainfall did not spatially and temporally agree well with observations. An obvious positive impact of ATMS radiances is effective for a short time. With the development of the atmospheric process, ATMS observations do not influence the main characteristics of the atmospheric system.

These conclusions are based on the heavy rainfall that occurred from 18 July to 20 July 2016, over North China. Further investigation on the impact of assimilating ATMS radiance observations on forecasts of the RMAPS-ST is required. However, conclusions from this study would be valuable for further understanding and application of ATMS radiances in regional NWP models, especially for heavy rainfall forecasts.

**Author Contributions:** Conceptualization, Y.X.; writing—original draft preparation, Y.X.; writing—review and editing, M.C., J.S., S.F., J.H., and Y.D.; funding acquisition, M.C. All authors have read and agreed to the published version of the manuscript.

**Funding:** This research was funded by the National Key Research and Development Program of China, grant number 2018YFC1506803.

**Acknowledgments:** We thank RMAPS-ST Authority for code support and sharing valuable data. We acknowledge National Centers for Environmental Prediction (NCEP) for the ATMS data.

**Conflicts of Interest:** The authors declare no conflict of interest.

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
