# Peer review of "Impacts of Assimilating ATMS Radiances on Heavy Rainfall Forecast in RMAPS-ST"

_remotesensing, doi:10.3390/rs12071147_

Round 1

Reviewer 1 Report

I am satisfied with the authors' effort to improve this manuscript. All my comments are amended properly. 

Author Response

Thanks for your comments and suggestions. We further improved the manuscript.

Reviewer 2 Report

Please find my comments in the attached document. 

Author Response

Thanks for the very constructive comments and suggestions. We have taken all the suggestions into consideration and revised the manuscript accordingly. The point-to-point responses have been given in the attachment.  

Reviewer 3 Report

The paper "Impacts of Assimilating ATMS Radiances on Strong Rainfall Forecast in RMAPS-ST", by Xie and colleagues presents a study on the assimilation of microwave radiances in a high resolution forecasting model to improve the quality of forecast in case of heavy rain. The validation is carried out on a 3-day episode occurred in northern China, and shows improvement of the overall prediction skills of the model in terms of commonly used performance indicators. Moreover, it provides more information on the vertical structure of features in the temperature and moisture profiles.

The paper is interesting and presents a widespread use of remote sensing data, it is fairly well written and deserves publication of Remote Sensing. Nevertheless, I suggest below few modification that in my opinion could improv the quality of the paper.

First, I suggest to refer to "heavy" rainfall instead of "strong" rainfall. 

In the formulas 1, 2 and 3, the comma at the end of the lines could confuse the reader, since it seems a "prime" symbol. I suggest to remove the comma.

The acronyms OMA and OMB should be spelled at their first appearance.

Some information on the instruments to measure rainfall rates and on the quality of the data should be added.

An information on the number of total samples for the different thresholds should be given. I guess that the class with rr>50 mm could be poorly populated, so that CSI it would not be the better indicator for this discussion. Try to use HSS instead, e.g. for figure 8, that is more "equitable".

One thing the Authors should explain is why they considered one single event: this results are for sure "event dependent" and more meaningful results would be obtained after the study of a wider event spectrum, as the Authors themselves notice at the end of the Conclusions (lines 489-490).  We have 9 years of NPP data, and the reader may wonder why only one case occurred 3.5 years ago is presented. I think that, at least, the reason for selecting this case should be more deeply explained.

Author Response

We first thank the very constructive comments from all the reviewers. We have taken all the comments and suggestions into consideration and revised the manuscript accordingly. All the changes have been highlighted in the revised manuscript. Our detailed responses are given as the  attachment.

Reviewer 4 Report

The paper by Yanhui Xie et al. is one of the best papers I have seen for the last six months.

My only complain is about of explanation about the assimilation process of ATMS data in the weather prediction system RMAPS-ST. I think the paper would beneficiate of a short Annex presenting both the RMAPS-ST weather prediction model and the assimilation process of ATMS data. The reader not specialist in the topic will appreciate a lot.

Author Response

We first thanks for the comments and suggestion. We have revised the manuscript accordingly. All the changes have been highlighted in the revised manuscript. Our detailed responses are given as the attachment.

This manuscript is a resubmission of an earlier submission. The following is a list of the peer review reports and author responses from that submission.

Round 1

Reviewer 1 Report

This manuscript to evaluate the impact of Advanced Technology Microwave Sounder (ATMS) radiance data assimilation on strong rainfall forecasts of heavy precipitation occurred over North China during 18-20 July 2016. Two evaluations; the initial conventional observations condition (based on the system of Rapid-refresh Multi-scale Analysis and Prediction System-Short-term (RMAPS-ST),) and the second after assimilating ATMS radiance observations on the forecast accuracy. And also forecast results from the two groups of experiments have been compared and evaluated against observations.

In my opinion, the scope of this study is clearly suited for the journal Remote Sensing. The analysis appears to have been carefully done and the statistical analysis appears to be appropriate. Some minor English issues do detract from the readability of the text. However, some outstanding issues must be addressed before the publication process can proceed.

In my opinion, the primary issue which must be resolved is the discussion section of this paper. It is not discussed appropriately, it looks like a result explanation not a discussion. The author barely discusses by comparing their results with previous studies. They did not give the possible reasons of having such results. They should also relate the performance assimilating ATMS radiance with the data retrieval method of ATMS.  The second major issue is the conclusion, It looks like a summery not a conclusion. It should be restated to a specific conclusion to the aspect of your Objective.    

Line 107 why you choose only “strong rainfall” what about little rainfalls? Is the prediction is more interested to rainstorm events.

Line 139 “The top pressure of the model is set to 50 hPa” why 50hPam or a reference?

Line 176 and  Line 178 “will be ” change to “was “ and do the same throughout your manuscript.  

Line 178 To consider observations from conventional sounding and ground-based stations as a reference for comparison, is the accuracy known? If yes please cite it.

Verification Strategy, It would be good if you include Heidke skill score (HSS) statistic which measures the accuracy of the estimates while accounting for matches due to random chances.

Line 214 Pearson correlation is oversensitive to extreme values (outliers) and insensitive to additive and proportional differences. Don’t you think this affects your result?? It would be preferable if you use Statistic metrics of Mean error, Mean absolute error and Nash-Sutcliffe efficiency coefficient.  

Reviewer 2 Report

Review of remotesensing-679599:

Impact of Assimilating ATMS Radiances on Forecasts of Strong Rainfall in RMAPS-ST

By

Yanhui Xie, Min Chen, Jiancheng Shi, Shuiyong Fan, Jing He and Youjun Dou

GENERAL COMMENTS: The paper analyzes the impact of the radiance assimilation from the Advanced Technology Microwave Sounder (ATMS) mounted on Suomi National Polar-orbiting Partnership (NPP) on the forecast of a severe event occurred over North China during 18-20 July 2016. The paper is sufficiently complete but a more accurate discussion of the results is required and in additional a careful check of the English.

SPECIFIC COMMENTS:

P 2, L 81: leading to a reduction

P 3, L 122-124: The authors don’t specify the version of the WRF model used. The resolution of the initial conditions? The frequency of the boundary conditions?

P 4, L 141-144: The cumulus scheme has been activated also in the inner domain?

P 6, L 197: false alarm rate (FAR).

P 7, L: 231: configuration is.

P 7, L 249 uncertainties

P 9, L 318-328: The authors should find the possible scientific explanations of these biases.

In general, I think that the discussion of the results is too poor. A more accurate discussion is necessary. The authors should try to find the links between the various results, trying to identify what are the atmospheric processes that lead to these behaviors.

Reviewer 3 Report

This study discusses the assimilation of Advanced Technology Microwave Sounder (ATMS) radiance data into the Rapid-refresh Multi-scale Analysis and Prediction System-Short-term (RMAPS-ST) model. The analysis is considered the strong rainfall events and investigating the impact of assimilation on the model forecast. The motivation of the study is discussed clearly in the manuscript. Also, the detailed description of the observations and precipitation products are included. The content of the paper and results are useful to researchers in the field, and it matches the requirements of publishing in this journal. I recommend accepting the article for publication in the present form.